# Single-photon oxidation of $C_{60}$ by self-sensitized singlet oxygen

Linqi Zhang[1], Chong Wang[2,3], Jiming Bao [2] & A. Kaan Kalkan [1✉]

$C_{60}$ is regarded as the most efficient singlet oxygen ($^1O_2$) photosensitizer. Yet, its oxidation by self-sensitized $^1O_2$ remains unclear. The literature hints both oxygen and $C_{60}$ must be at excited states to react, implying a two-photon process: first, oxygen is photosensitized ($^1C_{60} \bullet ^1O_2$); second, $C_{60}$ is photoexcited ($^1C_{60}^* \bullet ^1O_2$). However, this scheme is not plausible in a solvent, which would quench $^1O_2$ rapidly before the second photon is absorbed. Here, we uncover a single-photon oxidation mechanism via self-sensitized $^1O_2$ in solvents above an excitation energy of 3.7 eV. Using excitation spectroscopies and kinetics analysis, we deduce photoexcitation of a higher energy transient, $^3C_{60}^{**} \bullet ^3O_2$, converting to $^1C_{60}^* \bullet ^1O_2$. Such triplet-triplet annihilation, yielding two simultaneously-excited singlets, is unique. Additionally, rate constants derived from this study allow us to predict a $C_{60}$ half-life of about a minute in the atmosphere, possibly explaining the scarceness of $C_{60}$ in the environment.

[1] Functional Nanomaterials Laboratory, Oklahoma State University, Stillwater, OK 74078, USA. [2] Department of Electrical & Computer Engineering, University of Houston, Houston, TX 77204, USA. [3] Present address: School of Materials and Energy, Yunnan University, Kunming 650500 P.R., China. ✉email: kaan.kalkan@okstate.edu

Buckminsterfullerene ($C_{60}$), the most abundant fullerene as well as the most symmetric molecule in nature, has been studied extensively since its discovery in 1985[1]. Among the outstanding properties of $C_{60}$ are ability to accept up to six electrons, high intersystem crossing (ISC) quantum yield, and long-lived triplet states[2], which have stimulated a thriving research effort for applications in photovoltaics[3], photocatalysis[4], and molecular probes[5]. Additionally, the latter two attributes make $C_{60}$ an efficient singlet oxygen ($^1O_2$) sensitizer, particularly promising for photodynamic therapy and environmental remediation[6–8].

However, $C_{60}$ is subject to photodegradation in these applications at ambient temperature (while thermal oxidation of $C_{60}$ occurs at 370 K and above)[9,10]. The initial work on photooxidation (PO) of $C_{60}$ credited it to ozonation[11–13], which however is limited to excitation wavelengths shorter than 240 nm for photogeneration of $O_3$[14]. Later, however, another reactive oxygen species became the suspect, $^1O_2$[15–17]. To this end, the most seminal findings have been: (i) unless photoexcited, $C_{60}$ does not react with externally-produced $^1O_2$[16]; (ii) PO can occur under UV excitation (308 nm) in $O_2$ ambient (albeit formation of $^1O_2$ was not corroborated)[17]; and (iii) $C_{60}O$ is the major photoproduct[16–18]. Thus, the PO reaction is anticipated as $C_{60}^* + {}^1O_2 \rightarrow C_{60}O + \frac{1}{2}O_2$, where both $C_{60}$ and $O_2$ must be photoexcited. However, the mechanistic details of the photophysics and photochemistry remain unelucidated. A historical review of the $C_{60}$ PO literature is provided in Supplementary Note 1.

$C_{60}$ has long been known to be an efficient $^1O_2$ sensitizer. Yet, no evidence has been shown that $^1O_2$ reacts with its original $C_{60}$ sensitizer, which we refer to as "oxidation with self-sensitized $^1O_2$". Here, we present experimental evidence for this phenomenon. Although the lifetime of $^1O_2$, $\tau$, in air is exceptionally long (i.e., 45 min)[19], it shortens to microseconds to nanoseconds in solvents. Inspired by this broad range of $\tau$, we investigated PO of $C_{60}$ in hexane ($C_6H_{14}$), chloroform ($CHCl_3$), and carbon tetrachloride ($CCl_4$), where $\tau$ is 30, 207 and 87,000 µs, respectively[20]. Our kinetics study reveals $C_{60}$ concentration decays exponentially under UV excitation and the decay rate increases with $\tau$. We also show the decay dominantly occurs as a single-photon process above the photon energy ($h\nu$) threshold of 3.7 eV, being the onset of $1^1A_g \rightarrow 2^1H_u$ transition in $C_{60}$.

## Results and discussion

**Absorption spectroscopy.** Figure 1a, b shows time series optical absorption spectra for our slowest and fastest PO kinetics, which occur in $C_6H_{14}$ and $CCl_4$, respectively. The major $C_{60}$ absorption peaks at 256 and 328 nm are seen to decrease systematically, while the baseline rises indicative of a photoproduct, which is also evident from yellowing of the solution (Fig. 1b, inset). The spectrum of the excitation source (Fig. 1c) consists of a major narrow band peaking at 310 nm with no emissions below 250 nm. Hence, the possibility of $O_3$ generation is ruled out[14]. In Fig. 1b, the noise below 255 nm is due to the high absorption of $CCl_4$ attenuating the optical beam. However, it does not deteriorate the accuracy of the absorption peak of $C_{60}$ at 260 nm in $CCl_4$ (see Supplementary Discussion).

**Phosphorescence spectroscopy.** In Fig. 2a, the phosphorescence peak at 1273 nm substantiates photosensitization of $^1O_2$ by $C_{60}$ in the three solvents in our study. Figure 2b plots the measured phosphorescence peak intensity versus quenching rate constant ($k_q = 1/\tau$). Here, values of $\tau$ for the three solvents are borrowed from ref. [20] as listed above. The experimental data points match with the theoretical trend (Supplementary Eq. (48)). Hence, we confidently adopt the $\tau$ values from the literature.

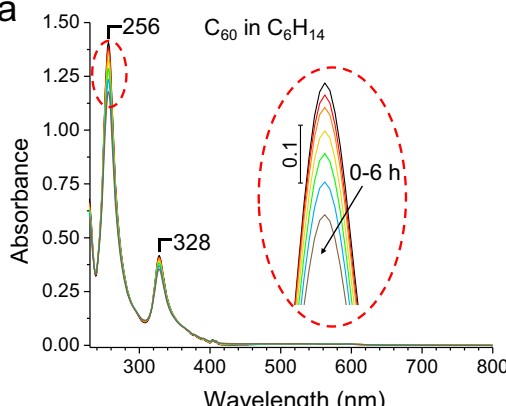

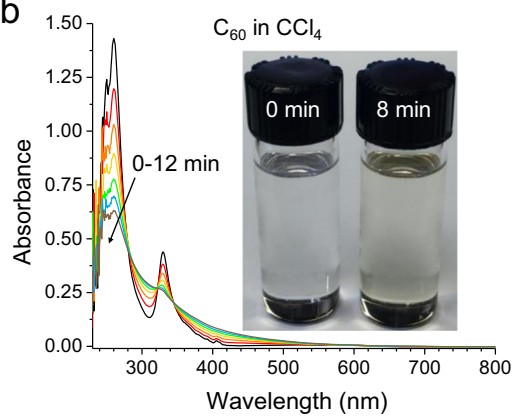

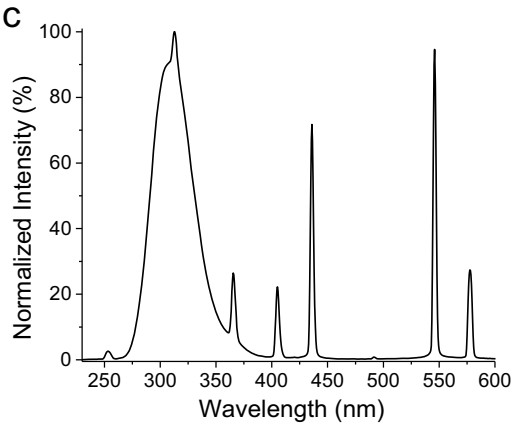

**Fig. 1 Time series absorbance spectra of $C_{60}$ under UV excitation of 3.74 mW/cm$^2$. a** In $C_6H_{14}$; **b** in $CCl_4$. The inset of (**b**) shows photos of $C_{60}$ solutions, unexposed and after 8 min of UV exposure. **c** The spectrum of the excitation source, UVP XX-15 UV lamp.

**Vibrational spectroscopy.** PO of $C_{60}$ is also characterized by the Fourier-transform infrared (FTIR) spectra in Fig. 3a, where C–O, C=O, and O–H stretching vibrations are indicative of $C_{60}$ oxidation[13]. We anticipate the O–H groups result from the Norrish type II reaction[21]. The evolution of C–H vibrational peaks suggest fragmentation of the $C_{60}$ cage subsequent to PO. The peak frequencies and important assignments are shown in Fig. 3b[22,23]. Detailed peak assignments are given in Supplementary Table 1.

**Mechanism of $C_{60}$ photooxidation in solvents.** At first, we are inclined to explain the oxidation of $C_{60}$ by its reaction with free

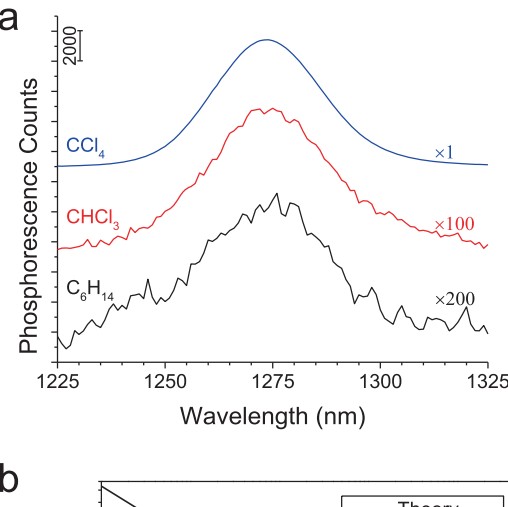

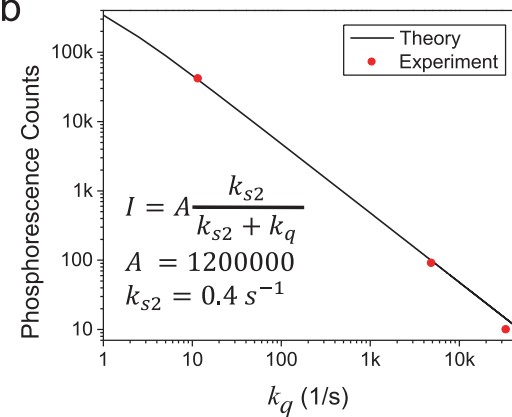

**Fig. 2 Phosphorescence of $^1O_2$ sensitized by $C_{60}$. a** Spectra in different solvents (under 375 nm radiation of 9.5 mW/cm$^2$ intensity). Here, [$C_{60}$] in $C_6H_{14}$ is 3.92 times higher than the usual concentration. **b** Match of theoretical intensity ($I$) with experiment (in (**a**)) validating the $k_q$ values adopted from the literature. $k_{s2}$ (0.4 s$^{-1}$) is the sensitization rate by $C_{60}$. The 2 in the subscript indicates $h\upsilon < 3.7$ eV (Supplementary Results).

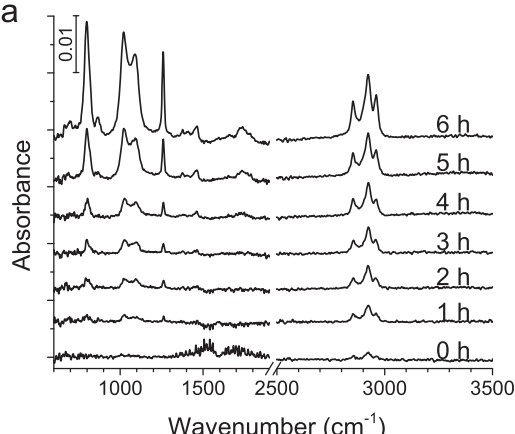

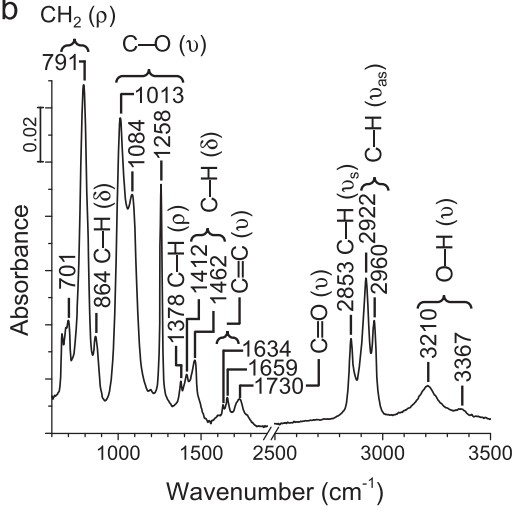

**Fig. 3 Time series FTIR spectra. a** $C_{60}$ in CHCl$_3$ under the same UV exposure conditions as in Fig. 1. **b** Assignment of FTIR peaks after 6 h of UV exposure (ρ: rocking; δ: bending; υ: stretching).

$^1O_2$. In this model, $^1O_2$ is released to the solvent after photo-sensitization by a $C_{60}$. Subsequently, it collides and reacts with a $C_{60}$ unless quenched by the solvent. This straightforward model (detailed in Supplementary Information; Kinetics Model) is consistent with our observation that PO rate increases with $\tau$. Additionally, its rate is quadratic in [$C_{60}$] as well as radiation intensity. On the contrary, the kinetics of $C_{60}$, as monitored from optical absorption (Fig. 4a) suggests exponential decay, i.e., $\frac{d}{dt}[C_{60}] = -k_{pd}[C_{60}]$, where $k_{pd}$ is the $C_{60}$ photodecay rate. Additionally, Fig. 4b establishes a linear dependence of $k_{pd}$ on excitation intensity, and hence a single-photon process. Thus, we rule out "oxidation with free $^1O_2$" as the dominant PO mechanism. Instead, consistent with the observed exponential decay, we propose "oxidation with self-sensitized $^1O_2$", where a $C_{60}$ molecule photosensitizes a $^1O_2$ and reacts with that same $^1O_2$ in a collision complex:

$$C_{60} + {}^3O_2 \xrightarrow{\text{collision}} C_{60}\bullet{}^3O_2 \xrightarrow{h\upsilon} C_{60}^*\bullet{}^3O_2 \xrightarrow{\text{sensitization}}$$
$$C_{60}\bullet{}^1O_2 \xrightarrow{\text{oxidation(?)}} C_{60}O \quad (1)$$

Yet, Scheme (1) has a flaw with the oxidation step, where $C_{60}$ and $^1O_2$ will not react, because $C_{60}$ must be excited to $C_{60}^*$. To meet this condition, a two-photon process could be proposed,

where the first photon excites $C_{60}$ to sensitize $^1O_2$ and the second one excites $C_{60}$ to a high energy singlet state, which thereafter reacts with $^1O_2$:

$$C_{60} + {}^3O_2 \xrightarrow{\text{collision}} C_{60}\bullet{}^3O_2 \xrightarrow{h\upsilon} C_{60}^*\bullet{}^3O_2 \xrightarrow{\text{sensitization}}$$
$$C_{60}\bullet{}^1O_2 \xrightarrow{h\upsilon} C_{60}^*\bullet{}^1O_2 \xrightarrow{\text{oxidation}} C_{60}O \quad (2)$$

However, a two-photon process is already excluded by our results (i.e., Fig. 4b). Additionally, Scheme (2) can be ruled out by fundamental considerations. Even the longest $\tau$ (0.087 s in CCl$_4$) is significantly shorter than the period between two subsequent excitations of $C_{60}$, being 3.7 s (see Supplementary Information; Kinetics Model). Therefore, before the second photon absorption occurs in Scheme (2), $C_{60}\bullet{}^1O_2$ will relax to $C_{60}\bullet{}^3O_2$ with a high probability. In other words, the excited $O_2$ and excited singlet $C_{60}$ will hardly coincide in time.

Accordingly, we are urged to consider a scheme, which allows simultaneous excitation of $O_2$ and $C_{60}$, after which they coexist and react. Scheme (1) considers the most basic photosensitization event, where $C_{60}$ returns to its ground singlet state after imparting its energy to $O_2$. On the other hand, it is possible that $C_{60}$ returns to an excited singlet state, $C_{60}^*$, (if it is photoexcited to a sufficiently high energy singlet state, $C_{60}^{**}$). Accordingly, Scheme

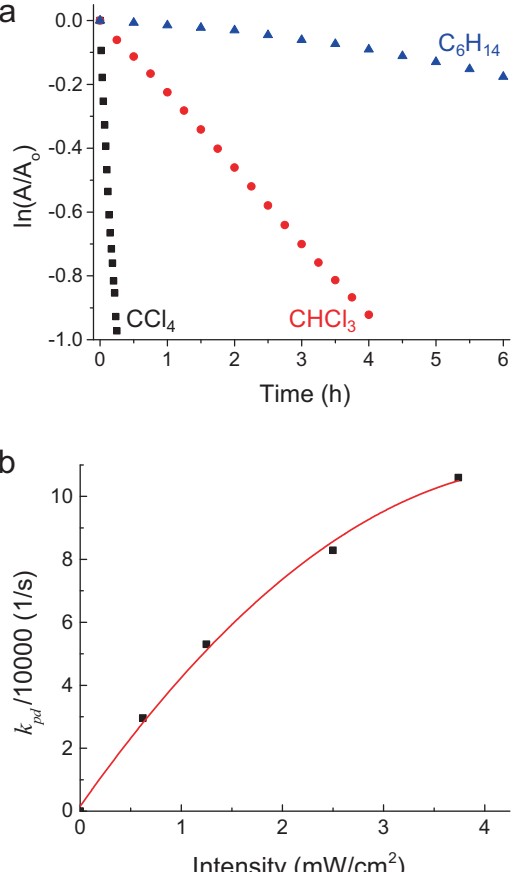

**Fig. 4 C$_{60}$ oxidation kinetics. a** C$_{60}$ time decay in different solvents. *A* is optical absorbance at 256 nm (peak), *A*$_o$ being the original value (prior to exposure). Although the photoproduct baseline is not subtracted, the slopes represent exponential decay rates (*k*$_{pd}$) with minimal error as corroborated in Supplementary Discussion. The UV exposures are same as in Fig. 1. **b** *k*$_{pd}$ in CCl$_4$ as a function of irradiation intensity.

(1) may be modified to:

$$C_{60} + {}^3O_2 \xrightarrow{collision} C_{60} \bullet {}^3O_2 \xrightarrow{h\upsilon} C_{60}^{**} \bullet {}^3O_2 \xrightarrow{sensitization}$$

$$C_{60}^* \bullet {}^1O_2 \xrightarrow{oxidation} C_{60}O \qquad (3)$$

The lowest energy $^1C_{60}^*$ is 2.33 eV above the ground state. Additionally, $^1O_2$ sensitization requires 0.98 eV while 0.37 eV is lost to exchange interaction during singlet-to-triplet conversion[24]. Therefore, a minimum excitation energy (*h*υ) of 3.68 eV is needed for Scheme (3) (Supplementary Fig. 4) to succeed. Consistently, our investigation using 455 and 395 nm LED excitations (2.73 and 3.14 eV) with similar photon-count exposures as in Fig. 1 yielded no detectable PO, although we confirmed $^1O_2$ sensitization for these excitations from the 1273 nm phosphorescence (Supplementary Fig. 1). In the below two paragraphs, we experimentally corroborate $1^1A_g \rightarrow 2^1H_u$ is the major driver of C$_{60}$ PO in the solvents. Interestingly, $1^1A_g \rightarrow 2^1H_u$ starts at 3.72 eV[25], being very close to the threshold energy for PO (Scheme (3)).

In Fig. 5a, excitation spectrum for $^1O_2$ phosphorescence (sensitization) closely follows C$_{60}$ absorption spectrum from 700 nm down to 370 nm. This trend is consistent with constant and near-unity $^1O_2$ photosensitization quantum yield by C$_{60}$, $\Phi_s$, as established in the literature[24]. Figure 5b shows the deconvolution

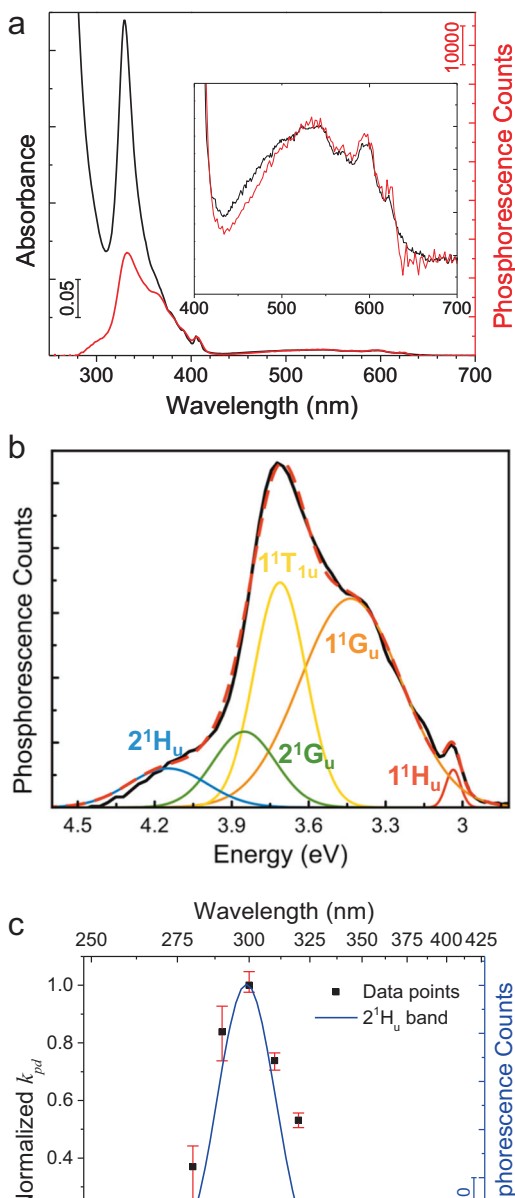

**Fig. 5 Excitation spectra. a** Overlay of the absorption spectrum of C$_{60}$ (black) and excitation spectrum for photosensitization of $^1O_2$ by C$_{60}$, monitored from $^1O_2$ phosphorescence counts at 1270 nm (red). **b** Deconvolution of the phosphorescence excitation spectrum. **c** Overlay of normalized *k*$_{pd}$ at different excitation wavelengths (black) and the deconvoluted 2$^1$H$_u$ band (blue). Confidence interval error bars are shown (red) after 3 independent measurements.

of the excitation spectrum to Gaussians below 400 nm. Each band marks an optical transition. Although these transitions may also be resolved from optical absorption, their deconvolution is more facile from our excitation spectrum.

Below 370 nm, however, $\Phi_s$ diverges from the absorption spectrum and drops significantly. On the other hand, the excitation spectrum for oxidation (i.e., *k*$_{pd}$) in Fig. 5c ("Methods") exhibits a reverse trend. The PO rate, *k*$_{pd}$, is essentially zero for

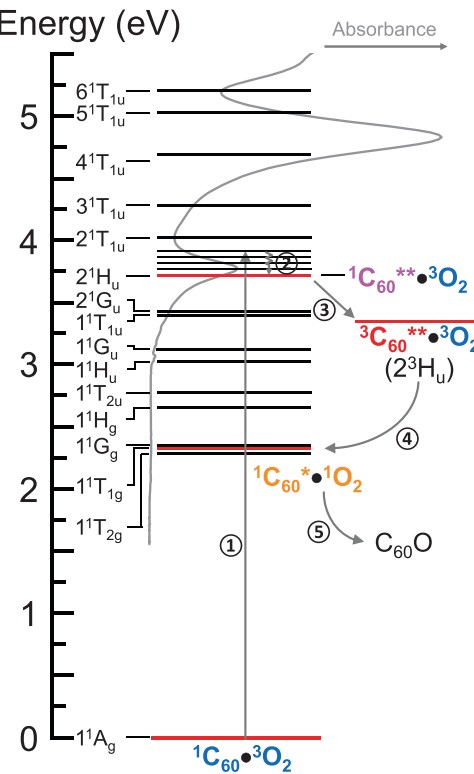

**Fig. 6 Jablonski diagram illustrating photooxidation of C$_{60}$.** To a first approximation, we adopt the energy structure of isolated C$_{60}$ for the C$_{60}$ of C$_{60}$•O$_2$. The gray curve represents the absorbance spectrum of C$_{60}$.

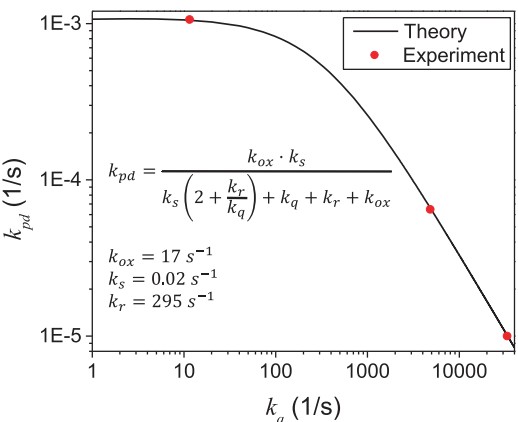

**Fig. 7 Fitting of theoretical $k_{pd}$ expression to experimental data.** The fitted values of $k_{ox}$ and $k_r$ are given in the inset.

$$k_{pd} = \frac{k_{ox} \cdot k_s}{k_s \left(2 + \frac{k_r}{k_q}\right) + k_q + k_r + k_{ox}}$$

$$k_{ox} = 17 \; s^{-1}$$
$$k_s = 0.02 \; s^{-1}$$
$$k_r = 295 \; s^{-1}$$

the spectral range, where $\Phi_s$ is at its maximum value of unity, but it is activated at the threshold of about 335 nm (3.70 eV), at which $\Phi_s$ is reduced to 0.37. Hence, the excitation trends in Fig. 5a, c, being spectrally different, underscore the fact that sensitization of $^1O_2$ is not sufficient for the oxidation of C$_{60}$. Specifically, as seen in Fig. 5c, the $k_{pd}$ spectrum matches the $2^1H_u$ band. These findings validate Scheme (3) as well as $1^1A_g \rightarrow 2^1H_u$ being the major driver of PO. Here, the normalized $k_{pd}$ values were derived from the $^1O_2$ phosphorescence intensity (i.e., counts proportional to [C$_{60}$]) kinetics (Supplementary Fig. 3).

While $1^1A_g \rightarrow 2^1H_u$ is the major driver of C$_{60}$ PO (Scheme (3)), $1^1G_u$, $1^1T_{1u}$, and $2^1G_u$ states can also be excited to their vibronic levels higher than 3.7 eV (from $1^1A_g$), as inferred from their deconvoluted bands in Fig. 5b. However, vibrational relaxation (VR) is the fastest process, quickly quenching $1^1G_u$, $1^1T_{1u}$, and $2^1G_u$ to their ground vibrational levels at 3.12, 3.40, and 3.43 eV, respectively (Fig. 6). Hence, ISC from these singlet states at above 3.7 eV is expected to be outcompeted by VR. Alternatively, PO (Scheme (3)) is possible from $1^1G_u$, $1^1T_{1u}$, and $2^1G_u$ vibronic states, if they transition to $2^1H_u$ by internal conversion (IC) before VR to below 3.7 eV. Because both IC and ISC are slower than VR by an order of magnitude or more, PO from $1^1G_u$, $1^1T_{1u}$, and $2^1G_u$ vibronic states will be minor, but may not be negligible. In conclusion, the major PO is expected to be through direct excitation of $2^1H_u$. Accordingly, $k_{pd}$ spectrum (data points) in Fig. 5c follows the $2^1H_u$ band. However, some deviation is seen, being highest for the 3.88 eV (320 nm) data point and over the $2^1H_u$ Gaussian, suggestive of additional excitations contributing, possibly through $1^1G_u$, $1^1T_{1u}$, and $2^1G_u$ as discussed above.

We illustrate Scheme (3) with the gray arrows in the Jablonski diagram of Fig. 6. First, C$_{60}$ is photoexcited through

$1^1A_g(^1C_{60}) \rightarrow 2^1H_u(^1C_{60}^{**})$. Then, $^1C_{60}^{**}$ transitions to $^3C_{60}^{**}$ ($2^1H_u \rightarrow 2^3H_u$) via ISC. Subsequently, triplet–triplet annihilation (TTA)[26] occurs with sensitization of $^1O_2$:$2^3H_u$•$^3O_2 \rightarrow 1^1T_{1g}$•$^1O_2$. TTA also leaves C$_{60}$ at an excited singlet state ($1^1T_{1g}$), which can readily react with $^1O_2$:$1^1T_{1g}$•$^1O_2 \rightarrow C_{60}O$. Hence, both $1^1T_{1g}$ and $^1O_2$, two energetic species, are created at the same time and same place (in collision complex) and have a higher chance to react. Another useful interpretation is that a single photon's energy ($h\nu$) is partially utilized in sensitizing $^1O_2$ while the excess energy leaves C$_{60}$ at an excited state, which can react with $^1O_2$. Scheme (3) may be expressed in more detail as:

$$^1C_{60} + {}^3O_2 \rightarrow {}^1C_{60} \bullet {}^3O_2 \xrightarrow{UV} {}^1C_{60}^{**} \bullet {}^3O_2 \xrightarrow{ISC}$$
$$^3C_{60}^{**} \bullet {}^3O_2 \xrightarrow{TTA} {}^1C_{60}^{*} \bullet {}^1O_2 \rightarrow C_{60}O \qquad (4)$$

We provide a mathematical analysis of PO kinetics for Scheme (3), which considers all the steps as well as reverse/competing processes, such as $^1O_2$ quenching, relaxation of $C_{60}^{*}$, and complex dissociations (see Supplementary Information; Kinetics Model). Despite the full complexity of this model, it predicts simply an exponential decay for [C$_{60}$], being consistent with the measured kinetics. The model allows us to write $k_{pd}$ as a function of $^1O_2$ quenching rate, $k_q = 1/\tau$. Fitting of this function to experimental data (Fig. 7) reveals the rate constants, $k_{ox}$ and $k_r$, associated with $C_{60}^{*} \bullet {}^1O_2 \xrightarrow{k_{ox}} C_{60}O$ and $C_{60}^{*} \bullet {}^1O_2 \xrightarrow{k_r} C_{60} \bullet {}^1O_2$, respectively, which has an interesting implication, as discussed below.

**Mechanism of C$_{60}$ photooxidation in the atmosphere.** PO of C$_{60}$ is expected to be significantly accelerated in the atmosphere thanks to dramatically prolonged $\tau$ in the air (i.e., tens of minutes). Indeed, PO can dominantly occur as a two-photon process (Scheme (2), illustrated in Supplementary Fig. 5) driven by visible and UVA photons, being abundant in solar radiation. Unlike in solvents, it is challenging to monitor PO of C$_{60}$ in air, since C$_{60}$ being at detectable concentrations in air, would quickly undergo aggregation as well as adsorption to enclosure walls. However, $k_{pd}$ can be predicted from the rate constants, $k_{ox}$ and $k_r$, which are already captured in the present work from C$_{60}$ dispersions in solvents. As such, we compute $k_{pd} = 0.011 \; s^{-1}$ for AM 1.5 solar radiation, suggesting a half-life of 63 s (see Supplementary Information; Kinetics Model). This rapid PO of C$_{60}$ in the atmosphere potentially explains its scarceness in the environment[27].

**Two excited singlets by triplet–triplet annihilation**. By its description in the literature, TTA involves Dexter energy transfer from a triplet to another, after which the acceptor transitions to a higher energy state (singlet), while the donor returns to its ground singlet state. On the other hand, Scheme (3) engages a unique TTA, which produces two excited singlets simultaneously and thereby enables an efficient photochemistry. This scheme is stimulating for the conception of novel efficient photochemical processes, implemented with $C_{60}$ or other photosensitizers.

## Methods

**$C_{60}$ solution preparation**. A stock solution of $C_{60}$ was first prepared by dissolving 1 mg of $C_{60}$ (Thermo Fisher Scientific, >99.9%) in 10 mL of solvent via ultra-sonication for 30 min. Next, the solution was kept undisturbed in the dark for 30 min to let insoluble aggregates (e.g., $C_{60}O$) settle down. Subsequently, the supernatant was transferred by a pipette to a spectrophotometer cell (optical path length of 10 mm) filled with the same solvent until the absorbance of $C_{60}$ at 256 nm reaches 1.42 (as monitored by a spectrophotometer), corresponding to $C_{60}$ concentration of 5.67 µM. Finally, the prepared solution was stored in a sealed glass vial and kept in the dark before use. High purity of $C_{60}$ in the prepared $C_{60}$ solution is confirmed by mass spectrometry (Supplementary Fig. 2).

**Excitation spectroscopy for photooxidation**. Photooxidation (photodecay) rate of $C_{60}$, $k_{pd}$, was measured as a function of excitation wavelength using the Fluorolog-3 spectrofluorometer. In a typical acquisition, 100 µL of $C_{60}$ in $CCl_4$ solution was placed in a standard microfluorescence cuvette (Science Outlet, 10 mm optical length, 0.35 mL capacity) and excited at selected wavelengths (from 250 to 420 nm, at 10 nm intervals) using a bandpass of 5 nm. The emission ($^1O_2$ phosphorescence) was parked at 1270 nm with a bandpass of 30 nm. At each excitation wavelength, the acquisition was performed 3 times and an unexposed sample was employed at each acquisition. The time-series phosphorescence intensity was collected in-situ at every 2 s with detector integration time of 2 s. Hence, the monochromatic optical beam of the spectrometer served as a dual probe simultaneously for measurement and exposure. For each excitation wavelength, $k_{pd}$ was derived from the exponential decay rate of the phosphorescence intensity, quantifying the decay rate of $C_{60}$ concentration (see Supplementary Fig. 3).

## Data availability
The data that support the findings of this study are available from the corresponding author upon request.

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

## Acknowledgements
The authors are indebted to Dr. David Jacob for his help with the Fluorolog Spectrometer in the Chemistry Undergraduate Teaching Laboratory. The authors also thank Dr. Steve Hartson of OSU Genomics and Proteomics Center for his assistance with mass spectrometry. A.K.K. acknowledges the funding by National Science Foundation (Award #1707008). J.B. acknowledges the support of Robert A. Welch Foundation (E-1728).

## Author contributions
L.Z. discovered the unexpectedly rapid photooxidation of $C_{60}$ in polystyrene. L.Z. and A.K.K. suspected the role of $^1O_2$ and decided to explore photooxidation of $C_{60}$ in solvents. A.K.K. hypothesized models, formulated the kinetics equations, and designed the experiments to explain the photooxidation mechanism. L.Z. prepared the samples, performed all the experiments, acquired all the reported data, and prepared all the figures. C.W. and J.B. repeated and confirmed the phosphorescence measurements (Fig. 2) with their custom-designed emission spectroscopy setup. L.Z. led the literature search. L.Z. and A.K.K. performed the data analysis. A.K.K. wrote the manuscript. A.K.K. and L.Z. wrote the Supplementary Information. The manuscript was checked and approved by all authors.

## Competing interests
The authors declare no competing interests.
