## [Peer Review File · Communications Chemistry]

Reviewers' comments:

Reviewer #1 (Remarks to the Author):

The topic of photooxidation of C60 and the underlying mechanism is very interesting.

The previous work is summarised well, but limited to C60 oxidation in solution. The later literature on photooxidation of C60 in solid form (or thin film) in air or O2 is not mentioned. (for instance the work of M. Wohlers, et al., e.g. Fullerene Science and Technology (1997) <http://dx.doi.org/10.1080/15363839708011973>) and Spring ACS (1996), and SyntheticMetals 77 (1996) 299).

The authors mention that C60O products may change their solubility and the solutions are left to rest to let these precipitate. How can then the authors be sure that they probe all of the photo-oxidized products in their absorption and IR spectroscopy measurements during exposure and that they can quantify the amount for product formed?

Can the authors conclude that the C60O product is epoxide, given that the IR spectra are showing increased absorption for a large variety of functional groups, C-H, CH2, O-H, C-O?

Why do the IR spectra of unexposed C60 not show the typical 4 IR peaks of C60, while they do show C-H vibrations in the 2800-3000 cm⁻¹ region?

Reviewer #2 (Remarks to the Author):

This paper reports and proposes a single-photon mechanism for photo-oxidation of C60 with a self-sensitized singlet-oxygen in solvents above 3.7 eV. The mechanism involves a triplet-triplet annihilation with the unusual formation of two singlet excited states (C60 and oxygen).

The topic is of great interest and the study appears to be carefully conducted and the paper provides convincing evidence for its conclusion. The novel mechanism proposed will influence thinking in the field and for these reasons I find the paper worth publishing.

I have no major issues with its content. The manuscript is mostly well written, and I only list few points that should be considered:

- 1) On page 1 the authors state that 'high intersystem crossing quantum yield is allowed by high symmetry' in C60. ISC has to do with spin-orbit coupling. What do the authors mean with symmetry?
- 2) Page 1, line 13 from top: collaborated should read corroborated, I suppose
- 3) Page 2, line 7 from bottom, "after photosensitized" should read "after photosensitization"
- 4) Page 4, line 2 from top: the assignment of the experimental transition in C60 to the 2Hu state is made here only on the basis of energy matching. However there are several states in C60 around that energy that could contribute to the absorption band. I don't think the authors can assess the symmetry of the transition they probe, therefore I would keep the discussion more general. There is for sure some plausible excited state in that energy region..

Reviewer #3 (Remarks to the Author):

Editorial Note: Please see attached pdf file for this Reviewer's "Remarks to the Author".

The paper reports the sensitization of $^3\text{O}_2$ by using the C_{60} molecule. The singlet second excited state was employed to convert $^3\text{O}_2$ to $^1\text{O}_2$ and thereby photooxidation of C_{60} . The authors concluded this based on the optical studies as well as the literature reports. However, the manuscript contains vague results and discussions. In fact, sometimes it is hard to follow the results and analysis. In conclusion, the manuscript warrants publication but major revision is needed. The revision should be along with the following points.

1. In Fig. 1a: Show the absorption changes using arrows pointing down or pointing up for easy to read
2. What is the origin of the peak of 1273 nm? Is it belonging to C_{60} or $^1\text{O}_2$? In Fig. 1c caption it was assigned to C_{60} . The 1273 nm corresponds to 0.98 eV energy. In the literature C_{60} , triplet energy is 1.62 eV (see J. Phys. Chem. 1991, 95, 11-12). The authors dealing with the second excited triplet state? If that is the case authors must discuss the quantum yield of this state, thereby the efficiency of the proposed photooxidation of C_{60}
3. FTIR: what is the source of O—H?
4. Fig 3b: Is it an excitation spectrum of C_{60} ? I don't think 1270 nm is the phosphorescence of C_{60} .
5. Page5: I am not sure what this sentence means...“We illustrate Scheme 3 with the violet arrows in the Jablonski diagram of Fig. 4. Following $1\text{Ag}^{\text{®}}$ 21Hu , 1C_{60}^{**} transitions to 23Hu by intersystem crossing (ISC)”

Authors' response to the Reviewers and description of the revisions

Manuscript COMMSCHEM-19-0338-T

"Hitting two molecules with one photon; photooxidation of C₆₀ by self-sensitized singlet oxygen"

We thank the reviewers for their meticulous review and constructive comments, according to which our manuscript was revised rigorously. Below is our response (in blue) and the description of our revisions (in green), addressing all the mistakes, weaknesses and questions pointed out by the reviewers. Additionally, we performed some voluntary revisions to enhance the rigor of our work and clarity of our presentation. These voluntary revisions are described at the end of this document. The revised text in our manuscript and figure captions are indicated in red.

Reviewer #1

The topic of photooxidation of C₆₀ and the underlying mechanism is very interesting.

1) The previous work is summarized well but limited to C₆₀ oxidation in solution. The later literature on photooxidation of C₆₀ in solid form (or thin film) in air or O₂ is not mentioned. (for instance the work of M. Wohlers, et al., e.g. *Fullerene Science and Technology* (1997) <http://dx.doi.org/10.1080/15363839708011973>) and *Spring ACS* (1996), and *Synthetic Metals* 77 (1996) 299).

The suggested references (*Fullerene Science and Technology*, **5**, 49 (1997) and *Synthetic Metals*, **77**, 299 (1996)) have carefully studied the thermally-activated reaction of O₂ with fullerenes (e.g., C₆₀ and C₇₀) at different temperatures. Critical temperatures for the initiation of oxidation (i.e., 370 K), cage opening (i.e., 470 K) and gasification (i.e., 570 K) were determined. Different oxidation products as well as structural changes (e.g., solid C₆₀ vs. polymerized C₆₀) at different elevated temperatures were revealed by UPS, XPS, XAS, FTIR and other advanced characterization tools.

Although the suggested references are concerned with thermal oxidation rather than photooxidation of C₆₀, they include important insights relevant to our work, especially in terms of identification of the oxidation products of C₆₀. Therefore, we cite the two suggested references as References 9 and 10 in our revised manuscript.

2) The authors mention that C₆₀O products may change their solubility and the solutions are left to rest to let these precipitate. How can then the authors be sure that they probe all of the photo-oxidized products in their absorption and IR spectroscopy measurements during exposure and that they can quantify the amount for product formed?

Before answering the Reviewer's question directly, let's review our purification process, which was performed only prior to photooxidation. This process was not employed after/during photooxidation at all. The C₆₀ employed in the present work is claimed to be high purity (Thermo Fisher Scientific, >99.9%). However, we were still concerned it might contain impurities, for example, C₆₀O. Therefore, we purified the C₆₀ before preparing the C₆₀ solution for photooxidation. In the purification step, we dissolved C₆₀ in a solvent by ultrasonication and then rested the solution. Because of the solubility difference, C₆₀ (dissolved in the solvent) was separated from the impurities (precipitates, albeit low), and the supernatant (higher purity C₆₀) was used in C₆₀ solution preparation. A minor precipitation took place during the purification step. However, after the C₆₀ solution was prepared, no further precipitation was observed (to the naked eye) in the purified stock solutions.

Second, we would like to answer the Reviewer's question, "How can then the authors be sure that they probe all of the photo-oxidized products in their absorption and IR spectroscopy measurements during exposure and that they can quantify the amount for

product formed?" Although C₆₀O is the major photooxidation product, our work does not rely on quantification of the C₆₀O concentration as a function of time. We only confirm C₆₀O as a photoproduct qualitatively using FTIR. Instead, our quantitative study focuses on the kinetics (decay) of C₆₀ by optical absorption, which forms the basis for testing different hypotheses/models and finally unraveling the correct mechanism. The formation of photoproducts does not change our results and conclusions as we focus on the decay of C₆₀. On the other hand, we have taken the optical absorption baseline due to photoproducts into consideration while extracting the photodecay rate constant of C₆₀ (Section G in the Supporting Information).

Additionally, we anticipate the aggregation of photoproducted C₆₀O to be negligible based on its low concentration in the solution. For example, we added an inset photo in Figure 1b during our revisions which shows pictures of C₆₀ in CCl₄ before and after photooxidation. The C₆₀ solution before photooxidation looks transparent and colorless, indicative of the lower concentration of C₆₀. The transmitted color of C₆₀ dissolved in CCl₄ (at higher concentrations, e.g., 1 mM) is magenta. During photooxidation, the color of the solution gradually turns to yellow (e.g., mainly due to C=O groups), indicating the formation of photoproducts. However, no precipitates were observed during the whole photooxidation process. Therefore, all species were suspended in the solvent and the optical absorption correctly recorded the baseline due to photoproducts. In addition, significant presence of suspended aggregates leading to Mie scattering is also ruled out, since the appearance of the solutions did not turn blurry as seen in the inset of Figure 1b. In conclusion, we believe all the C₆₀, as well as the photoproducts, are well suspended in the solution, and we probed all of the photo-oxidized products in our measurements.

3) Can the authors conclude that the C₆₀O product is an epoxide, given that the IR spectra are showing increased absorption for a large variety of functional groups, C–H, CH₂, O–H, C–O?

We thank the Reviewer for bringing up this important question. First, we deduce from FTIR that C₆₀O is not the only photoproduct of C₆₀. However, C₆₀O is the initial and most abundant photoproduct, which has been individually reported by four different groups (Reference 16: *Chem. Commun.*, 2493 (1998); Reference 17: *J. Chem. Soc. Commun.*, 2437 (1994); Reference 18: *J. Am. Chem. Soc.*, **114**, 1103 (1992); Reference S8: *J. Am. Chem. Soc.*, **113**, 5907 (1991)). In these works, C₆₀ in solvents was either exposed to ultraviolet (UV) lasers or radiation from a UV lamp (similar to our work), and the formation of C₆₀O was confirmed by high-performance liquid chromatography as well as mass spectrometry. On the other hand, we cannot ignore the formation of other C₆₀ oxides (e.g., C₆₀O_n (n=2 to 5)) reported in the literature (Reference 15: *J. Chem. Soc., Chem. Commun.*, 220 (1993); Reference S8: *J. Am. Chem. Soc.*, **113**, 5907 (1991)). Accordingly, in our manuscript (second paragraph, first page), we stated C₆₀O is the '**major**' photoproduct instead of the 'only' photoproduct.

Second, we have not called C₆₀O an "C₆₀ epoxide", because the way oxygen bonds to C₆₀ has not been established certainly in the literature, although the most likely structure has been claimed to be the epoxide (Reference 18: *J. Am. Chem. Soc.*, **114**, 1103 (1992)). Therefore, in our manuscript, we simply refer to it as "C₆₀O" instead of "C₆₀ epoxide".

Third, formation of different functional groups (shown in our FTIR spectra) during C₆₀ photooxidation is in agreement with the literature (Reference 13: *Nature*, **351**, 277 (1991)). As aforementioned, the major photoproduct of C₆₀ is C₆₀O. However, the C₆₀O subsequently undergoes photooxidation as well, which eventually leads to fragmentation of fullerene cage and formation of various functional groups. Our work mainly focuses on the initial (1st step) photooxidation of C₆₀ where it transforms to C₆₀O. Exploring the

photodegradation of $C_{60}O$ as well as other C_{60} photoproducts would be very interesting, but it is the subject of a future investigation.

Finally, the formation of photoproducts does not change our results and conclusions as we focus on the decay of C_{60} . On the other hand, we have taken the optical absorption baseline due to photoproducts into consideration while extracting the photodecay rate constant of C_{60} (Section G in the Supporting Information).

4) Why do the IR spectra of unexposed C_{60} not show the typical 4 IR peaks of C_{60} , while they do show C–H vibrations in the 2800-3000 cm^{-1} region?

First, we would like to review the performance of our ATR-FTIR spectrometer. We have obtained the FTIR spectra of **bulk** C_{60} using both diamond and germanium (Ge) ATR crystals (Figure A). The 4 allowed T_{1u} vibrations of C_{60} at 523, 573, 1182 and 1429 cm^{-1} are seen, which agree with the literature (*J. Phys Chem*, 96, 4262 (1992)). The presence of negative peaks of C_{60} in Figure A (diamond crystal) is an artifact, resulting from the similar refractive index of bulk C_{60} (2.0 as reported by *Opt. Express*, **26**, 27441 (2018)) and diamond (2.4 as reported by *J. Opt. Soc. Am. B*, **32**, 1718 (2015)) in IR. This artifact disappears when we change the ATR crystal to Ge (refractive index of 4.4 as reported by *ECS Transactions*, **69**, 279 (2015)). However, Ge crystal has two major drawbacks which adversely affect our results: 1) spectral cutoff at 550 cm^{-1} , masking the two most intense peaks of C_{60} at 523 and 573 cm^{-1} . 2) lower signal-to-noise ratio by 20 times compared to the diamond crystal. Attenuation of the signal is due to shorter penetration depth into the sample because of higher refractive index mismatch (i.e., between C_{60} and Ge). Therefore, the IR peaks of the C_{60} photoproducts can be measured with higher signal-to-noise with the diamond crystal. Accordingly, we employed a diamond crystal in our investigation.

Figure A. ATR-FTIR spectra of the bulk C_{60} .

Second, we would like to answer the question, “Why do the IR spectra of unexposed C_{60} not show the typical 4 IR peaks of C_{60} ?” As described in the Experimental Procedures Section of the Supporting Information, C_{60} sample for the ATR-FTIR acquisition was prepared by drying 10 μL of C_{60} solution on the ATR diamond crystal. Given the low concentration of C_{60} in the solution (i.e., 5.67 μM) and low volume (i.e., 10 μL) applied, the amount of C_{60} on the diamond crystal is limited. This low quantity limits the detection of the 4 IR peaks of C_{60} . If

we had increased the quantity of C_{60} on the ATR crystal towards bulk quantities, the 4 IR peaks would have been captured, as shown in Figure A.

Third, we would like to answer the question, "Why does the IR spectrum of unexposed C_{60} show C–H vibrations in the $2800\text{--}3000\text{ cm}^{-1}$ region?" The short answer is that the C–H vibrations are not associated with the unexposed C_{60} , because the C–H peak intensities do not increase when IR spectrum is acquired for the bulk C_{60} (as shown in Figure A). On the other hand, the presence of C–H vibrations in the IR background spectrum is known in the literature. Figure B shows a representative IR background spectrum taken in reference to air (CH 362 Experimental Chemistry I, FTIR Spectroscopy, Oregon State University: <https://chemistry.oregonstate.edu/courses/ch361-464/ch362/irinstrs.htm>), and vibrational peaks of the atmosphere components (e.g., H_2O and CO_2) and C–H are shown.

Figure B. A representative IR background spectrum.

In a typical ATR-FTIR measurement, we first acquire the background spectrum (i.e., no sample on the ATR-crystal, which is in full contact with air) and then subtract it from the spectrum of C_{60} . The subtraction is done automatically by our Bruker FTIR spectrometer. It is likely that, during the drying of the solvent (CCl_4) on the ATR crystal, hydrocarbons from the ambient air is dissolved in the solvent and then coated on the crystal upon drying. Such an organic layer may lead to the observation of residual C–H vibrations.

As the Reviewer may question, once the C–H vibrations cannot be rid by background subtraction, then H_2O and CO_2 peaks should also appear in the C_{60} spectrum. In fact, we did observe such vibrations in our spectra. For example, in Figure 2a of our manuscript, there are features at around 1500 cm^{-1} , which we attribute to H_2O .

To further validate our answer to the Reviewer, this time using Ge crystal (instead of diamond), we acquired an FTIR spectrum of pure CCl_4 solvent dried on the Ge crystal (no C–H vibrations in CCl_4). The Ge crystal was cleaned in a sequence by acetone, IPA and deionized water using nonwoven cleanroom wipers. Then a background spectrum was obtained with the crystal in contact with air only. Subsequently, $100\text{ }\mu\text{L}$ of CCl_4 solvent was spotted on the Ge detector and left to dry. A stain pattern can be seen on the Ge crystal after CCl_4 is completely evaporated. Then, the spectrum of this residual layer was acquired. Figure C shows the FTIR spectrum of the residual layer after background subtraction. We see C–H vibrations in $2800\text{--}3000\text{ cm}^{-1}$ again. Hence, we infer the C–H vibrations originate from the ambient hydrocarbons dissolved in CCl_4 during drying.

Figure C. ATR-FTIR spectrum of the residual layer, formed after spotting and drying of 100 μL CCl_4 on the Ge ATR crystal.

Reviewer #2

This paper reports and proposes a single-photon mechanism for photo-oxidation of C_{60} with a self-sensitized singlet-oxygen in solvents above 3.7 eV. The mechanism involves a triplet-triplet annihilation with the unusual formation of two singlet excited states (C_{60} and oxygen). The topic is of great interest and the study appears to be carefully conducted and the paper provides convincing evidence for its conclusion. The novel mechanism proposed will influence thinking in the field and for these reasons, I find the paper worth publishing. I have no major issues with its content. The manuscript is mostly well written, and I only list a few points that should be considered:

1) On page 1 the authors state that 'high intersystem crossing quantum yield is allowed by high symmetry' in C_{60} . ISC has to do with spin-orbit coupling. What do the authors mean with symmetry?

We are sorry for causing this confusion. We did not mean ISC is driven/induced by high symmetry. YES, ISC is driven/induced by spin-orbit coupling. We attempted to mean symmetry '**enhances**' the ISC quantum yield, but we admit our wording may be confusing. To avoid such confusion, we revised and shortened this statement to: "high intersystem crossing (ISC) quantum yield".

On the other hand, to understand the high ISC quantum yield of C_{60} , one must really consider the effect of symmetry (closely related to geometry and spherical shape of C_{60}). C_{60} has 60 rotational symmetries and a symmetry order of 120. C_{60} is uniquely the only molecule which belongs to icosahedral (I_h) symmetry. Although we may include a clear and thorough explanation in our manuscript, such background information will lengthen the text. Yet, mentioning "symmetry" briefly will potentially confuse readers. Therefore, we have decided to just mention the high ISC quantum yield in the said statement without mentioning the attribute of symmetry. However, we would like to provide the following explanation to the Reviewer.

In C_{60} , ISC quantum yield is high because of two reasons, being both related to 'high symmetry in C_{60} '. YES, ISC is definitely driven/induced by spin-orbit coupling. However,

spin-orbit coupling in C₆₀ is enhanced by its 'high symmetry' as mentioned in our original manuscript (Reference S18: *Electron Transfer I. Topics in Current Chemistry*, **169**, 348, (1994); *J. Am. Chem. Soc.*, **113**, 2780 (1991)). Hence, ISC rate constant is increased.

Second, this high symmetry leads to highly symmetric ground and excited states in C₆₀ in terms of spatial distribution of electron wavefunction and its phase. For example, as seen from Figure 4 of our manuscript, the ground state (1¹A_g) and first four excited states (1¹T_{2g}, 1¹T_{1g}, 1¹G_g and 1¹H_g) have the same (gerade or even) symmetry. Hence, electric dipole transitions are forbidden (Laporte rule) between the ground state and these four excited states of C₆₀. The transition rates are not zero due to disruption of symmetry by asymmetric vibrations (vibronic coupling), but the rates are still low enough to result in long lifetimes for excited singlet states. Once the excited singlet state is longer-lived, it has higher chance for ISC to a triplet state, because the singlet state can be longer acted (perturbed) by the spin-orbit coupling Hamiltonian.

2) Page 1, line 13 from top: collaborated should read corroborated, I suppose.

Corrected – We thank the Reviewer!

3) Page 2, line 7 from the bottom, "after photosensitized" should read "after photosensitization".

Corrected – We thank the Reviewer!

4) Page 4, line 2 from top: the assignment of the experimental transition in C₆₀ to the 2¹H_u state is made here only on the basis of energy matching. However, there are several states in C₆₀ around that energy that could contribute to the absorption band. I don't think the authors can assess the symmetry of the transition they probe; therefore, I would keep the discussion more general. There is for sure some plausible excited state in that energy region.

We thank the Reviewer for bringing up this very important question. We agree with the Reviewer. Accordingly, we revised the discussion, making it less restrictive. To address this point, we made four revisions. First, we changed the wording "... 1¹A_g → 2¹H_u being the driver of PO" to "... 1¹A_g → 2¹H_u being the **major** driver of PO ..." in two sentences.

Second, we removed the sentence, "In C₆₀, the transition just above this energy threshold is 1¹A_g → 2¹H_u starting at 3.72 eV (333 nm), as established computationally.²⁵" As the Reviewer pointed out, this sentence was wrong, because there are other transitions above 3.7 eV. However, it contained some correct information, too, which we moved to the end of the paragraph as: "Interestingly, 1¹A_g → 2¹H_u starts at 3.72 eV,²⁵ being very close to the threshold energy for PO."

Third, and most importantly, we added the following paragraph:

"While 1¹A_g → 2¹H_u is the major driver of C₆₀ PO (Scheme 3), 1¹G_u, 1¹T_{1u} and 2¹G_u states can also be excited to their vibronic levels higher than 3.7 eV (from 1¹A_g), as inferred from their deconvoluted bands in the inset of Fig. 3b. However, vibrational relaxation (VR) is the fastest process, quickly quenching 1¹G_u, 1¹T_{1u} and 2¹G_u to their ground vibrational levels at 3.12, 3.40 and 3.43 eV, respectively (Fig. 4). Hence, ISC from these singlet states at above 3.7 eV is expected to be outcompeted by VR. Alternatively, PO (Scheme 3) is possible from 1¹G_u, 1¹T_{1u} and 2¹G_u vibronic states, if they transition to 2¹H_u by internal conversion (IC) before VR to below 3.7 eV. Because both IC and ISC are slower than VR by an order of magnitude or more, PO from 1¹G_u, 1¹T_{1u} and 2¹G_u vibronic states will be minor, but may not be negligible. In conclusion, the major PO is expected to be through direct excitation of 2¹H_u. Accordingly, *k_{pd}* spectrum (data points) in Fig. 3c follows the 2¹H_u band. However, some deviation is seen, being highest for the 3.88 eV (320 nm) data point and above the

2^1H_u Gaussian, suggestive of additional excitations contributing, possibly through 1^1G_u , 1^1T_{1u} and 2^1G_u as discussed above.”

Fourth, in agreement with the Reviewer’s comment, we did not fit the k_{pd} spectrum to a Gaussian (which was the case in the original manuscript). Hence, we removed the red Gaussian in the original Figure 3c. The remaining blue Gaussian is not a fit to the k_{pd} data points. The blue curve shows the 2^1H_u Gaussian, imported from Figure 3b (deconvoluted from the excitation spectrum for photosensitization of 1O_2 by C_{60}).

Additionally, during our revisions, we repeated the measurement associated with Figure 3c three more times and we added error bars.

We believe these revisions should address the Reviewer’s concerns. However, we would also like to add the following discussion for further clarification.

Please, note that our assignment (i.e., $1^1A_g \rightarrow 2^1H_u$ is the transition responsible for photooxidation of C_{60}) is not only on the basis of energy matching. More importantly, as shown in Figure 3c, the 2^1H_u peak matches with k_{pd} (oxidation rate constant) spectrum, extracted from carefully conducted kinetics measurements. Accordingly, based on two evidences, we concluded that the $1^1A_g \rightarrow 2^1H_u$ transition is the **major** driver for C_{60} photooxidation instead of other transitions.

We absolutely agree there are multiple different transitions at 3.7 eV and above. As a reminder, the lowest excited singlet C_{60} ($^1C_{60}^*$) has energy of 2.33 eV above the ground state. Additionally, 1O_2 sensitization requires 0.98 eV while about 0.37 eV is lost to exchange interaction during singlet-to-triplet conversion. Therefore, a minimum excitation energy of $3.68 \approx 3.7$ eV is needed for Scheme 3 to succeed. For example, as seen from the inset of Figure 3b, a 3.8 or 3.9 eV photon can excite the ground state C_{60} to 1^1G_u , 1^1T_{1u} and 2^1G_u vibronic states (in addition to 2^1H_u). In the inset of Figure 3b, the 1^1G_u , 1^1T_{1u} and 2^1G_u bands and other bands are deconvoluted from the excitation spectrum of 1O_2 photosensitization (as monitored from 1O_2 phosphorescence counts). Therefore, these bands not only mark different transitions (optical absorption by C_{60}), but they also sensitize 1O_2 . Assignment to these absorption bands are made according to the literature (Reference 25: *J. Chem. Phys.*, **97**, 6496 (1992)). Therefore, a 3.9 eV photon, for example, can result in 4 different transitions and all of these 4 transitions can sensitize 1O_2 (although not all at the same quantum yield). However, the critical question is: is oxidation of C_{60} possible with all of these 4 transitions at 3.9 eV?

To answer this question, we need to consider the dynamics of the above excited states. C_{60} at these excited states is expected to relax through vibrational relaxation (VR) and internal conversion (IC). It is well established that relaxation of excited C_{60} by fluorescence is very low probability (even from its first excited singlet state – a fact explained by high symmetry in C_{60}). Characteristic times for VR is 10^{-12} - 10^{-10} s (Bernard Valeur, *Molecular Fluorescence Principles and Applications*) and it occurs faster than IC (10^{-11} - 10^{-9} s) and much faster than ISC (10^{-10} - 10^{-8} s). Therefore, in the majority of the excitations, the excited states will first relax to their ground vibrational levels before any ISC occurs. However, except for 2^1H_u , all other states (1^1G_u , 1^1T_{1u} and 2^1G_u) at their ground vibrational levels are below 3.7 eV in energy. Hence, no photooxidation of C_{60} will occur from ground vibrational levels of 1^1G_u , 1^1T_{1u} and 2^1G_u (through Scheme 3) although a 3.9 eV photon can excite these states and ISC can occur from their ground vibrational levels leading to 1O_2 sensitization. In other words, $^1C_{60}^* \bullet ^1O_2$ cannot form from ground vibrational levels of 1^1G_u , 1^1T_{1u} and 2^1G_u . (only $^1C_{60} \bullet ^1O_2$ will form).

Nevertheless, exact ISC as well as IC rates from the 1^1G_u , 1^1T_{1u} , 2^1G_u vibronic states are not known. Given the high ISC rate of $\sim 10^9$ s $^{-1}$ (Reference 24: *J. Phys. Chem.*, **95**, 11 (1991))

from the first excited state of C_{60} , the ISC rates from 1^1G_u , 1^1T_{1u} , 2^1G_u at above 3.7 eV may not be negligible, although they are expected to be low probability against VR.

Additionally, it is possible, albeit with low probability, that the excited states, 1^1G_u , 1^1T_{1u} and 2^1G_u may transition to 2^1H_u by IC before they vibrationally relax to below 3.7 eV (i.e., below ground vibrational level of 2^1H_u as well as threshold energy for PO (Scheme 3)). If this IC to 2^1H_u occurs, photooxidation of C_{60} may occur as if 2^1H_u is excited. However, IC is roughly an order of magnitude slower than vibrational relaxation and the probability of this event is expected to be on the order of 0.1 or less. In general, like ISC, IC also takes place from the ground vibrational levels due to VR being the fastest process. Indeed, the relaxation dynamics we discuss here is also responsible for Kasha's rule.

Reviewer #3

The paper reports the sensitization of 3O_2 by using the C_{60} molecule. The singlet second excited state was employed to convert 3O_2 to 1O_2 and thereby photooxidation of C_{60} . The authors concluded this based on the optical studies as well as the literature reports. However, the manuscript contains vague results and discussions. In fact, sometimes it is hard to follow the results and analysis. In conclusion, the manuscript warrants publication but major revision is needed. The revision should be along with the following points.

1) In Fig. 1a: Show the absorption changes using arrows pointing down or pointing up for easy to read.

The figure has been revised accordingly. We thank the Reviewer.

2) What is the origin of the peak of 1273 nm? Is it belonging to C_{60} or 1O_2 ? In Fig. 1c caption it was assigned to C_{60} . The 1273 nm corresponds to 0.98 eV energy. In the literature C_{60} , triplet energy is 1.62 eV (see J. Phys. Chem. 1991, 95, 11-12). The authors dealing with the second excited triplet state? If that is the case authors must discuss the quantum yield of this state, thereby the efficiency of the proposed photooxidation of C_{60} .

We apologize for being careless. The samples are C_{60} , but of course the peak at 1273 nm originates from 1O_2 phosphorescence. In Fig. 1c, we meant to say "Phosphorescence spectra of 1O_2 , sensitized by C_{60} , in different solvents (under 375 nm radiation of 9.5 mW/cm² intensity)." The caption has been corrected in our revised manuscript accordingly.

In our work, we are dealing with the 2^3H_u excited triplet state of C_{60} , which is clearly depicted in the C_{60} energy structure diagram of Figure 4. It converts from the 2^1H_u singlet state via intersystem crossing (ISC). 2^1H_u is the 10th excited singlet state of C_{60} (Figure 4).

ISC from 2^1H_u to 2^3H_u is expected to be efficient due to two reasons: 1) large spin-orbit coupling caused by the nearly spherical shape of C_{60} (Reference S18: *Electron Transfer I. Topics in Current Chemistry*, **169**, 348, (1994); *J. Am. Chem. Soc.*, **113**, 2780 (1991)); 2) dipole-forbidden transition from the 2^1H_u state to lower excited singlet states due to the same state symmetry (i.e., ungerade). HOWEVER, the quantum yield for 1O_2 sensitization is not necessarily high from 2^1H_u (or more directly from 2^3H_u). Indeed, using the data of Figure 3b, we compute this quantum yield for 1O_2 sensitization as 10%, as described below.

In Figure 3b, excitation spectrum for 1O_2 phosphorescence (at 1270 nm, generated by excitation of C_{60}) is shown in red, and the absorption spectrum of C_{60} is shown in black. As established in Reference S7 (*J. Phys. Chem.*, **95**, 11 (1991)), the quantum yield for sensitization of 1O_2 by C_{60} (Φ_s) is close to unity in the visible range. Therefore, in Figure 3b, we overlap the red and black curves at the visible range (excitation spectrum for 1O_2 phosphorescence and absorption spectrum of C_{60}), so the ratio of the height of the red curve to that black curve at any wavelength represents Φ_s . As discussed in our original manuscript, Φ_s drops significantly at wavelengths shorter than 370 nm. Indeed, in our

original manuscript, we had reported Φ_s as 0.37 at 335 nm (3.70 eV) as contributed from multiple excited states.

Presently, we do not know the reason, why Φ_s systematically drops for wavelengths shorter than 370 nm. However, if C_{60} did not undergo ISC after excitation at these wavelengths, it would relax to its lower excited singlet states, and finally to its first excited singlet state, from which ISC and sensitization would occur at close to 100% quantum yield, anyways. Hence, Φ_s would be again close to unity. Therefore, decrease in Φ_s is not due to lower ISC rates at these excited states. At this point, we can only speculate, that the reason for falling of Φ_s with increasing excitation energy is the detachment of 3O_2 from C_{60} . At higher excited states, the excited electron is accommodated in antibonding orbitals, which are farther out from the carbon atoms. This excited electron (charge) distribution possibly interferes with 3O_2 adsorbed to the surface, lowering its affinity to C_{60} .

As such, Φ_s drops to 10% at 299 nm, being the peak of the 2^1H_u band, as shown in the inset of Figure 3b. This is the overall Φ_s at 299 nm, which is total number of 1O_2 sensitized, divided by total number of photons absorbed. Additionally, as reported by Reference S17 (*Chem. Phys. Lett.*, **593**, 72 (2014)), the optical absorption of C_{60} at 299 nm is essentially contributed by the 2^1H_u alone. Therefore, the quantum yield for 1O_2 sensitization by 2^1H_u is computed approximately as 10%.

However, this number (i.e., 10%) is not the quantum yield for photooxidation. The sensitized 1O_2 does not undergo reaction with C_{60}^* at 100% efficiency. The probability of the oxidation reaction for the $^1C_{60}^* \cdot ^1O_2$ complex is $\frac{k_{ox}}{k_q + k_r + k_{ox}}$, where k 's are the rate constants for oxidation, 1O_2 quenching and $^1C_{60}^*$ relaxation. Since k_q is solvent dependent, so is this probability, being computed as 0.053 for CCl_4 . Hence, as questioned by the Reviewer, finally we can compute the quantum yield for oxidation of C_{60} by excitation to 2^1H_u state as: $10\% \times 0.053 = 0.53\%$. We included this computation in Section L of the Supporting Information.

3) FTIR: what is the source of O—H?

We thank the Reviewer for bringing up this important question. Various groups, such as C—H, C—O, C=C and most importantly C=O, are identified by our FTIR measurements. These findings agree with the literature (Reference 13: *Nature*, **351**, 277 (1991)). The presence of various groups suggests the fragmentation of C_{60} cage after extensive oxidation (e.g., 6 h exposure of C_{60} in $CHCl_3$). The fragmentation of C_{60} cage after extensive oxidation is also reported in the literature (Reference 9: *Fullerene Science and Technology*, **5:1**, 49 (1997); Reference 10: *Synthetic Metals*, **77**, 299 (1996); Reference 13: *Nature*, **351**, 277 (1991)). We infer the O—H groups could result from the Norrish type II reaction (Reference 21: *Macromolecules*, **1**, 98 (1968)) of the produced carbonyl (C=O) moieties in the fragmented C_{60} . The reaction mechanism (N. J. Turro, V. Ramamurthy, and J. C. Scaiano, *Principles of Molecular Photochemistry: An Introduction*, **Chapter 1**, 16 (2009)) can be generalized in Fig. D. Per Reviewer's inquiry, we have decided to add a sentence to our revised manuscript: "We anticipate the O—H groups result from the Norrish type II reaction.²¹".

Figure D. Norrish type II reaction.

4) Fig 3b: Is it an excitation spectrum of C₆₀? I don't think 1270 nm is the phosphorescence of C₆₀.

The Reviewer is correct. We apologize for being careless. The samples are C₆₀, but of course the peak at 1270 nm is associated with the ¹O₂ phosphorescence. The caption has been corrected in our revised figure to: "b) Overlay of absorption spectrum of C₆₀ (black) and excitation spectrum for photosensitization of ¹O₂ by C₆₀, monitored from ¹O₂ phosphorescence at 1270 nm (red)."

5) Page5: I am not sure what this sentence means. "We illustrate Scheme 3 with the violet arrows in the Jablonski diagram of Fig. 4. Following $1^1A_g \rightarrow 2^1H_u$, $^1C_{60}^{**}$ transitions to 2^3H_u by intersystem crossing (ISC)".

In Figure 4, we show the steps (transitions) involved in Scheme 3 using a Jablonski diagram. Scheme 3 is the novel pathway for photooxidation of C₆₀O, which is the focus of our submitted work. The major objective of our manuscript is to elucidate Scheme 3. Scheme 3 is described by text and symbols in our manuscript, but, in our opinion, its illustration using a Jablonski diagram in Figure 4 provides enhanced clarity. As different from a typical Jablonski diagram, we also show the reaction of $^1C_{60}^*$ with ¹O₂, producing C₆₀O, so that the Scheme 3 is described completely in Figure 4. The electronic/vibrational/chemical steps or processes are indicated by 5 arrows (violet-colored).

While performing our investigation, we needed an extended energy diagram showing the electronic states of C₆₀. However, the energy diagrams available in the literature were in limited energy ranges and not to-scale. Hence, we decided to construct a complete and precise-to-scale diagram from the tabulated energies of the electronic states. We also superposed the absorption spectrum of C₆₀ on our diagram, so one can associate the optical absorption bands (transitions) clearly with certain electronic states. We believe Figure 4 will be a facilitating resource for future investigations on optical/electronic properties of C₆₀.

In Figure 4, C₆₀ in the 1^1A_g state is the ground state. It is also denoted by $^1C_{60}$ (1 in the superscript denotes a singlet state). C₆₀ in the 1^1T_{1g} excited state is denoted by $^1C_{60}^*$ (the * indicates the first excited state). C₆₀ in the 2^1H_u excited state is denoted by $^1C_{60}^{**}$ (the ** indicates a higher energy excited state). C₆₀ in the 2^3H_u excited state is denoted by $^3C_{60}^{**}$ (the 3 in the superscript denotes a triplet state).

The sentence, "Following $1^1A_g \rightarrow 2^1H_u$, $^1C_{60}^{**}$ transitions to 2^3H_u by intersystem crossing (ISC)," describes the following processes: C₆₀ in the ground state (1^1A_g) absorbs a photon and is excited to the 2^1H_u state. Then, the C₆₀ in the 2^1H_u state ($^1C_{60}^{**}$) transitions to the 2^3H_u state by intersystem crossing (singlet-to-triplet transition). Since this sentence is questioned by the Reviewer, we felt the need to revise it for better clarity as follows: "First, C₆₀ is photoexcited through 1^1A_g ($^1C_{60}$) \rightarrow 2^1H_u ($^1C_{60}^{**}$). Then, the $^1C_{60}^{**}$ transitions to $^3C_{60}^{**}$ ($2^1H_u \rightarrow 2^3H_u$) via ISC."

Voluntary revisions by the authors

1) While the present work elucidates an original photooxidation mechanism for C₆₀, it also discloses for the first time photooxidation of C₆₀ in solvents. As we document, this photooxidation can be as quick as 46% decay of C₆₀ in 8 minutes under 3.74 mW/cm² of UV radiation, when CCl₄ is used as the solvent. This radiation intensity is significantly lower than that of solar radiation (~100 mW/cm²), but it is concentrated in the UVB region. To make this point more evident, we added inset photos in Figure 1b during our revisions. These photos show C₆₀ in CCl₄ before and after 8 min of UVB exposure (3.74 mW/cm²). Remarkably, the colorless C₆₀ solution turns yellow in just 8 minutes. Accordingly, we also added text to a sentence as: "The major C₆₀ absorption peaks at 256 and 328 nm are seen

to decrease systematically, while the baseline rises indicative of a photoproduct, which is also evident from yellowing of the solution (Fig. 1b inset)."

2) We revised Figure 3b by replacing the old data of the red curve (excitation spectrum for photosensitization of $^1\text{O}_2$ by C_{60} , monitored from $^1\text{O}_2$ phosphorescence at 1270 nm) with new data. The old data was acquired using a quartz optical cell with a window width of 10 mm, which was broader than the width of the optical excitation beam. Hence, during the acquisition of the old data, the excitation beam partially exposed the C_{60} solution, only at its center. In the acquisition of the new data, we used a quartz cell with a window width of 1 mm and the height of the solution column was set to 1 cm. Hence, the C_{60} solution was entirely enclosed by the exciting beam. In result, the old and new data differed in terms of better reproducibility of the new data in the UVC range. The higher irreproducibility in the old acquisition could be due to diffusion and convection effects between the exposed and unexposed regions of the solution. Additionally, in the new data, the 2^1T_u band (highest energy band in the old data) is essentially negligible. In this range, the excitation beam intensity is weaker leading to lower signal-to-noise and division of the sample signal intensity by the reference beam intensity may cause artifacts due to low value in the denominator. We confirmed disappearance of the 2^1T_u band in the new data is not due to an artifact due to faster photodegradation of C_{60} . We monitored the time decay of phosphorescence at this wavelength range carefully. The emission intensity was two orders of magnitude lower compared with the 2^1H_u band, but it was steady-state and no time decay of the 2^1T_u signal was recorded. Therefore, we have adopted the new data and revised Figure 3b with the new data. The Supporting Information has also been revised for the different optical cell employed.

3) During our revisions, we also reproduced the data of Figure 3c three more times and we also plotted the data with error bars in the revised Figure 3c.

4) In the revised manuscript, we added the sentence: "In Fig. 1b, the noise below 255 nm is due to high absorption of CCl_4 attenuating the optical beam. However, it does not deteriorate the accuracy of the absorption peak of C_{60} at 260 nm (SI)." This statement was corroborated by additional measurements disclosed in Section M of the Supporting Information. Additionally, as mentioned above, we took some of our discussions in this document (response to the Reviewers) to the Supporting Information. We added Section L.

5) We have made few and minor changes with the wording, tenses, and definite/indefinite articles (the/a). The changes were shown in red.

REVIEWERS' COMMENTS:

Reviewer #1 (Remarks to the Author):

The authors have made a serious effort to respond to the concerns I raised and have completed the manuscript with clarifying information and have carried out additional experiments to answer the questions posed by the reviewers.

In my opinion the revised manuscript can be published in its present form.

Reviewer #2 (Remarks to the Author):

The authors revised the paper with great attention. They replied carefully to all the observations and suggestions requested. I personally remember having seen very few replies so nicely and carefully carried out. Therefore I am perfectly satisfied by the reply of the authors and by the amendment of the manuscript which is therefore publishable in the present form.

Reviewer #3 (Remarks to the Author):

I am satisfied with the revised manuscript. The authors addressed the raised points satisfactorily.